# SGD ON NEURAL NETWORKS LEARNS ROBUST FEATURES BEFORE NON-ROBUST

## ABSTRACT

Neural networks are known to be vulnerable to adversarial attacks - small, imperceptible perturbations that cause the network to misclassify an input. A recent line of work attempts to explain this behavior by positing the existence of non-robust features - well-generalizing but brittle features present in the data distribution that are learned by the network and can be perturbed to cause misclassification.

In this paper, we look at the dynamics of neural network training through the perspective of robust and non-robust features. We find that there are two very distinct "pathways" that neural network training can follow, depending on the hyperparameters used. In the first pathway, the network initially learns only *predictive*, robust features and *weakly predictive* non-robust features, and subsequently learns predictive, non-robust features. On the other hand, a network trained via the second pathway eschews predictive non-robust features altogether, and rapidly overfits the training data. We provide strong empirical evidence to corroborate this hypothesis, as well as theoretical analysis in a simplified setting. Key to our analysis is a better understanding of the relationship between predictive non-robust features and adversarial transferability. We present our findings in light of other recent results on the evolution of inductive biases learned by neural networks over the course of training.

Finally, we digress to show that rather than being "quirks" of the data distribution, predictive non-robust features might actually occur across datasets with different distributions drawn from independent sources, indicating that they perhaps possess some meaning in terms of human semantics.

## 1    INTRODUCTION

Neural networks have achieved state of the art performance on tasks spanning an array of domains like computer vision, translation, speech recognition, robotics, and playing board games (Krizhevsky et al. (2012); Vaswani et al. (2017); Graves et al. (2013); Silver et al. (2016)). However in recent years, their vulnerability to adversarial attacks - small, targeted input perturbations, has come under sharp focus (Szegedy et al. (2013); Papernot et al. (2017); Carlini & Wagner (2017); Athalye et al. (2018); Schmidt et al. (2018)).

Ilyas et al. (2019) propose that neural network vulnerability is at least partly due to neural networks learning *well-generalizing* but *brittle* features that are properties of the data distribution. From this point of view, an adversarial example would be constructed by modifying an input of one class slightly such that it takes on the non-robust features of another class.

They provide empirical evidence for their theory by training a model on adversarially perturbed examples labeled as the *target class*, and showing that this model generalizes well to the original, unperturbed distribution.

Another unrelated line of work (Brutzkus et al. (2018); Ji & Telgarsky (2019); Li & Liang (2018)) aims to study the properties of the functions learned by gradient descent over the course of training. Nakkiran et al. (2019) and Mangalam & Prabhu (2019) independently showed that Stochastic Gradient Descent (SGD) learns simple, almost linear functions to start out, but then learns more complex functions as training progresses. Li et al. (2019) showed that models trained with a low learning rate learn easy-to-generalize but hard-to-fit features first, and thus perform poorly on easy-to-fit patterns.

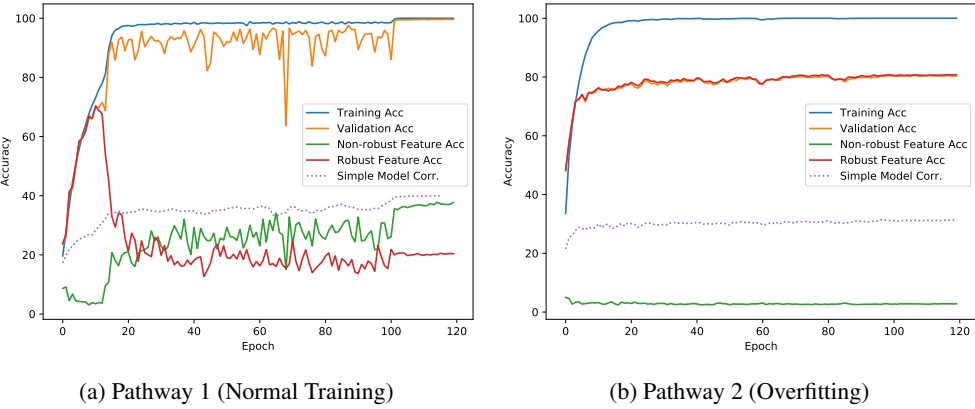

(a) Pathway 1 (Normal Training)                    (b) Pathway 2 (Overfitting)

Figure 1: (Best viewed in color). Neural network training follows two very different pathways based on the choices of hyperparameters. These are training graphs of two Resnet50 models trained on relabeled CIFAR-10 adversarial examples. See Section 4 for more details.

In this paper, we study gradient descent on neural networks from the perspective of robust and non-robust features. Our main thesis is that based on choices of hyperparameters, neural network training follows one of two pathways, :

- **Pathway 1 (Informal) :** The neural network first learns *predictive* robust features and *weakly predictive* non-robust features. As training progresses, it learns predictive *non-robust* features, and having learned both robust and non-robust predictive features, achieves good performance on held-out data. This is the pathway that Ilyas et al. (2019) used to prove their theory.

- **Pathway 2 (Informal) :** The neural network learns *predictive* robust features and *weakly predictive* non-robust features (as in Pathway 1). But thereafter, it begins to fit the *noise* in the training set, and quickly achieves zero training error. In this scenario, the network learns only the *robust* predictive features and shows modest generalization on held-out data.

Through a series of experiments, we validate our two-pathway hypothesis, investigate the specific circumstances under which Pathway 1 and Pathway 2 occur, and analyze some properties of the two pathways. We will also develop a closer understanding of the relationship between adversarial transfer and *predictive* non-robust features, which will aid our analysis of the two pathways.

The rest of this paper is organized as follows. Section 2 sets up the notation and definitions we use. In Section 3, we investigate the link between adversarial features and transferability. In Section 4 we provide empirical evidence for the two-pathway hypothesis and analyze some characteristics of each pathway. Section 5 presents a theoretical analysis of gradient descent on a toy linear model. We show that for different choices of initial parameters, the linear model exhibits properties of the first and second pathways. We digress to explore whether non-robust features can occur *across datasets* in Section 6, and discuss future research directions in Section 7.

## 2 DEFINITIONS AND PRELIMINARIES

Consider the binary classification setting, where $\mathcal{D}$ is a joint distribution over the input space $\mathcal{X}$ and the labels $\{-1, 1\}$ [1]. In this setting, Ilyas et al. (2019) define a *feature* as any function $f : \mathcal{X} \to \mathbb{R}$, scaled such that $\mathbb{E}_{(x,y)\in\mathcal{D}}[f(x)] = 0$ and $\mathbb{E}_{(x,y)\in\mathcal{D}}[f(x)^2] = 1$. A feature is said to be $\rho$-*useful* if

$$\mathbb{E}_{(x,y)\in\mathcal{D}}[y \cdot f(x)] > \rho \tag{1}$$

for some $\rho > 0$, and $\gamma$-*robust* if

$$\mathbb{E}_{(x,y)\in\mathcal{D}}\left[\inf_{\delta\in\Delta(x)} y \cdot f(x+\delta)\right] > \gamma \tag{2}$$

---

[1] This framework can easily be adapted to the multi-class setting

for some $\gamma > 0$ and some family of perturbations $\Delta$. For brevity, we sometimes refer to a $\rho$-useful, $\gamma$-robust feature ($\rho, \gamma > 0$) simply as a *robust feature*. Let $\rho_{\mathcal{D}}(f)$ be the largest $\rho$ for which $f$ is $\rho$-useful. A feature $f$ is said to be (highly) predictive or weakly predictive if $\rho_{\mathcal{D}}(f)$ is high or low respectively.

A *useful, non-robust feature* as defined by Ilyas et al. (2019) is one that is $\rho$-useful for some $\rho > 0$, but is not $\gamma$-robust for any $\gamma > 0$. They propose the following experiment to demonstrate the existence of these features.

Let $C$ be a classifier trained with Empirical Risk Minimization (ERM) on an empirical distribution $\widehat{\mathcal{D}}$. We operate under the following assumption.

**Assumption 1.** *If a distribution $\mathcal{D}$ contains a useful feature, then a classifier $C$ trained with ERM on an empirical distribution $\widehat{\mathcal{D}}$ drawn from $\mathcal{D}$ will learn this feature, provided that we avoid finite sample overfitting through appropriate measures such as regularization and cross-validation.*

Let $L_C(x, t)$ denote the loss of $C$ on input $x$, for a target label $t$. Construct adversarial examples by solving the following optimization problem :

$$x_{adv} = \underset{\|x - x'\| \leq \epsilon}{\arg\min} \, L_C(x', t) \tag{3}$$

In particular, construct a distribution called $\widehat{\mathcal{D}}_{det}$ comprised of $(x_{adv}, t)$ pairs by using Equation 3 with $t$ chosen deterministically according to $y$ for each $(x, y) \in \widehat{\mathcal{D}}$. In the binary classification setting, $t$ must be $-y$, so

$$\mathbb{E}_{(x_{adv}, t) \in \widehat{\mathcal{D}}_{det}} \, [t \cdot f(x_{adv})] > 0, \quad \text{if } f \text{ is non-robustly useful under } \mathcal{D} \tag{4}$$

$$\mathbb{E}_{(x_{adv}, t) \in \widehat{\mathcal{D}}_{det}} [-t \cdot f(x_{adv})] > 0, \quad \text{if } f \text{ is robustly useful under } \mathcal{D} \tag{5}$$

It is observed that a neural network trained on $\widehat{\mathcal{D}}_{det}$ achieves non-trivial generalization to the *original* test set, that is $\mathcal{D}$. From this, we can conclude that non-robust features exist and are useful for classification in the normal setting.

**Remark :** Goh (2019b) showed that the $\widehat{\mathcal{D}}_{rand}$ dataset constructed by choosing $t$ randomly in the above procedure, suffers from a sort of "robust feature leakage". PGD introduces faint robust cues in the generated adversarial example that can be learned by the model. But on the $\mathcal{D}_{det}$ dataset, the robust features are correlated with a deterministic label which is different from $t$. Hence we use the $\widehat{\mathcal{D}}_{det}$ dataset in preference to the $\widehat{\mathcal{D}}_{rand}$ for all our experiments.

**Two kinds of non-robust features :** Goh (2019a) points out a subtle flaw with the above definition of a non-robust feature - highly predictive non-robust features can arise from "contamination" of a robust feature with a non-robust feature, instead of something meaningful. To see how this can happen, consider a *highly predictive* robust feature $f_R$ and a *weakly predictive* non-robust feature $f_{NR}$. Let $f_C$ be a "contaminated" feature that is a simple sum of $f_R$ and $f_{NR}$ (appropriately normalized). Then it is possible to construct a scenario in which

$$\mathbb{E}[y \cdot f_R(x)] > 0 \qquad\qquad \mathbb{E}\left[\inf_{\delta \in \Delta(x)} y \cdot f_R(x + \delta)\right] > 0 \tag{6}$$

$$\mathbb{E}[y \cdot f_{NR}(x)] \gtrsim 0 \qquad\qquad \mathbb{E}\left[\inf_{\delta \in \Delta(x)} y \cdot f_{NR}(x + \delta)\right] \ll 0 \tag{7}$$

$$\mathbb{E}[y \cdot f_C(x)] > 0 \qquad\qquad \mathbb{E}\left[\inf_{\delta \in \Delta(x)} y \cdot f_C(x + \delta)\right] < 0 \tag{8}$$

$f_C$ is thus a highly predictive non-robust feature. Now when you train a model on $(x + \delta, -y)$ pairs, $f_C = f_R + f_{NR}$ is still correlated with $-y$. But $f_{C'} = -f_R + f_{NR}$ is more correlated, so the model will learn this combination in preference to $f_C$ and will not generalize on the original distribution. In fact, thanks to learning $-f_R$, it will generalize to the distribution with flipped labels, i.e., $y \rightarrow -y$. In our analysis and experiments, when we refer to non-robust features, we will exclude such contaminated features.

**Illustrative Example :** Consider a dataset of dog and cat images, where most dog images have snouts and most cats do not have snouts. Most cats have slightly lighter eyes than dogs, and making

| Name | Feature | Type of Feature |
|:---:|:---:|:---:|
| $f_1$ | Snout $\implies$ 1, otherwise $-1$ | Predictive Robust |
| $f_2$ | Dark Eyes $\implies$ 1, otherwise $-1$ | Predictive Non-Robust |
| $f_3$ | First pixel is an odd number $\implies$ 1, otherwise $-1$ | Weakly Predictive Non-Robust |
| $f_4$ | $f_1 + f_3$ | Contaminated Robust |

Table 1: An example illustrating the different kinds of features. Dogs and cats are labeled $+1$ and $-1$ respectively. Most dogs have dark eyes and snouts. A small majority of dog images start with an odd numbered pixel value.

the eyes slightly darker or lighter is part of the set of valid adversarial perturbations. Suppose that a very small majority of the dog images start with a pixel that has an odd numbered value. Then the different types of features in this dataset are enumerated in Table 1.

For $f_2$, $(x + \delta, -y)$ pairs would be dogs with lighter eyes, labeled as cats. The network trained on these examples will learn Snout $\implies$ Cat, Light Eyes $\implies$ Cat.

Since the eye-color is predictive of the true label, the second feature will ensure that the neural network has non-trivial performance on the original distribution. This is what Ilyas et al. (2019) observed in their experiments. $f_2$ is thus a true non-robust feature.

For $f_4$, $(x+\delta, -y)$ pairs would be dog images with the first pixel value converted to an even number, labeled as cats. The network trained on these examples will learn Snout $\implies$ Cat, Dark Eyes $\implies$ Cat, and Even Pixel $\implies$ Cat.

None of these will be particularly helpful on the true distribution, but the first two will be useful on the *flipped* distribution, i.e., where dogs are relabeled as cats. $f_4$ is thus a contaminated robust feature, and not a non-robust feature.

**Remark :** A network that learns only robust features but with contaminants can still be very vulnerable to adversarial attacks, as the above example shows. The weakly predictive non-robust feature $f_3$ can be manipulated to consistently cause misclassification on out-of-distribution inputs.

## 3 NON-ROBUST FEATURES AND TRANSFERABILITY

The phenomenon of adversarial transferability (Papernot et al., 2016), where a non-trivial fraction of the adversarial examples generated for one neural network are still adversarial to other neural networks trained independently on the same data, can be readily explained in terms of non-robust features.

By Assumption 1, different neural networks trained using ERM on a distribution would learn the predictive non-robust features (like Dark Eyes $\implies$ Dog) present in the distribution. One would then construct an adversarial example by modifying an input such that the predictive non-robust features flip (modify all dog images to have lighter eyes). Then this adversarial example would transfer to all the different networks that have learned to rely on the non-robust features.

A natural question to ask is, does *all* adversarial transferability arise from predictive non-robust features? Nakkiran (2019) showed that by explicitly penalizing transferability during PGD, one can construct adversarial examples that do not transfer, and from which it is not possible to learn a generalizing model. This establishes that adversarial examples that do not transfer, do not contain predictive non-robust features.

Here we provide a simpler experiment that constructs non-transferable adversarial examples without explicitly penalizing transferability. This experiment also establishes a stronger claim, that adversarial examples transfer *if and only if* they exploit predictive non-robust features.

Let the CIFAR-10 dataset form the data distribution $\mathcal{D}$. Train two Resnet50 models (He et al., 2016) on $\widehat{\mathcal{D}}$ and ensure by Assumption 1 that both networks have learned the predictive non-robust features of the distribution by using regularization and cross-validation across a grid of hyperparameters.

Construct a $\widehat{\mathcal{D}}_{det}$ dataset for the first network using Equation 3 where $t$ is chosen deterministically according to $y$ using the transformation $t = (y+1)\%10$. We use Projected Gradient Descent (PGD) (Madry et al., 2018) to solve the optimization problem in Equation 3. Split the adversarial examples into two categories - those that transfer to the second network *with* their target labels, and those that do not. Relabel all adversarial examples $x_{adv}$ with their target label $t$, and train a Resnet50 model on $(x_{adv}, t)$ pairs from each category.

As Equations 4 and 5 suggest, for $(x_{adv}, t) \sim \widehat{\mathcal{D}}_{det}$, the non-robust features of $\mathcal{D}$ are predictive of $t$, but the robust features of $\mathcal{D}$ are predictive of $(t-1)\%10$. So if a neural network trained on a subset of $\widehat{\mathcal{D}}_{det}$ learns predictive non-robust features, it will generalize to $\mathcal{D}$, and if it learns predictive robust features, it will generalize to the *shifted* distribution $\mathcal{D}_{shift}$ :

$$\mathcal{D}_{shift} = \{(x, (y+1)\%10) \ : \ (x,y) \sim \mathcal{D}\} \quad (9)$$

Figure 2 shows the performance of these two networks on $\mathcal{D}$ and $\mathcal{D}_{shift}$. We can see that the network trained on the examples that transfer generalizes well to $\mathcal{D}$, but the network trained on the examples that do not transfer generalizes to $\mathcal{D}_{shift}$. The configuration in the figure is as a result of a thorough grid search over hyperparameters with the metric for success being performance on $\mathcal{D}$.

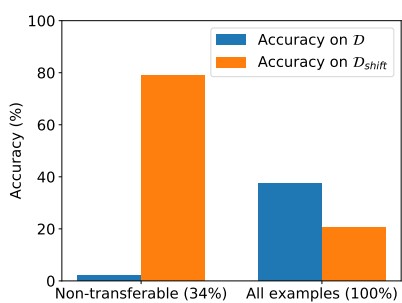

Figure 2: Clean accuracy of a Resnet50 model trained on subsets of *relabeled* adversarial examples

Along with Assumption 1, our experiment establishes that the examples that transfer contain predictive non-robust features, and the examples that don't transfer don't contain predictive non-robust features. In particular, we claim the following :

**Claim 1.** *Train two networks $N_1$ and $N_2$ on a common dataset such that both networks learn the predictive non-robust features present in the dataset. Then an adversarial example generated for $N_1$ transfers to the second network if and only if this example contains predictive non-robust features.*

Further, if a neural network $C$ has learned predictive non-robust features, then PGD will construct some adversarial examples with predictive non-robust features (see Equation 4), and vice-versa. This allows us to infer the following property, which we will use in our analysis in the next section :

**Claim 2.** *If a neural network $N_2$ has learned the predictive non-robust features in a dataset, then adversarial examples generated for another network $N_1$ using PGD will transfer to $N_2$ if and only if $N_1$ has also learned predictive non-robust features.*

## 4 THE TWO PATHWAY HYPOTHESIS

### 4.1 EXPERIMENTAL SETUP

We use the CIFAR-10 training set as our empirical distribution $\widehat{\mathcal{D}}$, and train a neural network $N_1$ using ERM on $\widehat{\mathcal{D}}$ with cross-validation and regularization such that it learns non-robust features by Assumption 1. Construct the $\mathcal{D}_{det}$ dataset according to the procedure described in Section 2, where the reference model $C$ is $N_1$ and the adversarial target $t$ is chosen deterministically as $t = (y+1)\%10$. Split $\mathcal{D}_{det}$ into training and validation sets and train a new neural network $N_2$ on the training set.

As we discussed in Section 3, if $N_2$ is able to generalize to $\mathcal{D}$, then $N_2$ must have learned the *predictive non-robust* features of $\mathcal{D}$, and if $N_2$ is able to generalize to $\mathcal{D}_{shift}$, then $N_2$ must have learned the *predictive robust* features of $\mathcal{D}$. This is depicted in Figure 3 in the context of our illustrative example from Section 2.

We use the accuracy on $\mathcal{D}$ (respectively, $\mathcal{D}_{shift}$) as a proxy for how much of the model's performance can be attributed to its learning predictive non-robust (respectively, robust) features. We refer to these as "non-robust feature accuracy" and "robust feature accuracy".

Finally, the accuracies on the training and validation splits of $\mathcal{D}_{det}$ tell us how well the model has *fit the training data*, and whether the model is *overfitting*. We train the network $N_2$ using SGD for

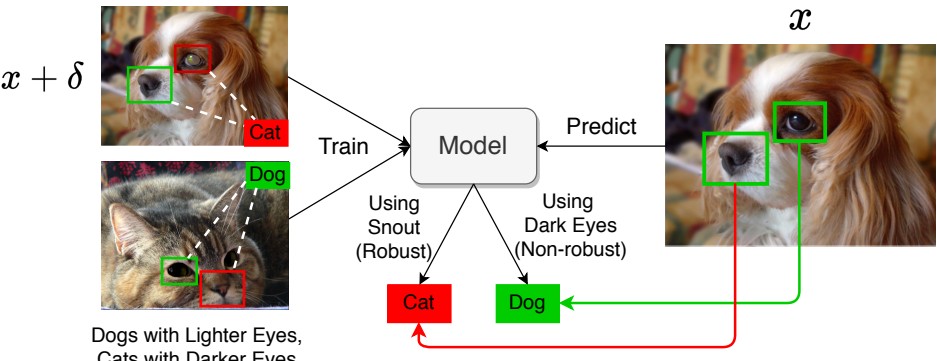

Figure 3: Train a model on dogs with light eyes labeled as cats, and cats with dark eyes labeled as dogs. If the model classifies a clean dog image as a cat, then it must have learned the predictive robust feature (snout), and if it classifies a dog as a dog, it must have learned the predictive non-robust feature (dark eyes).

120 epochs using different combinations of learning rate and regularization, and plot the evolution of these four metrics over the course of training.

## 4.2 RESULTS AND DISCUSSION

We observe that training follows two very distinct regimes or pathways depending on the choice of hyperparameters.

The **first pathway**, illustrated in Figure 1a occurs when the model is trained with regularization of some sort - either in the form of a high initial learning rate (LR), $L_2$ weight decay, or data augmentation. The model starts out by learning only predictive robust features (possibly with contaminants), but at some point switches to learning a combination of robust and non-robust predictive features. Training and validation accuracy steadily increase, and the model ends with both training and validation accuracy close to 100%.

The **second pathway**, illustrated in Figure 1b occurs when the model is trained with a low starting learning rate, little or no $L_2$ weight decay and no data augmentation. The model starts out similar to the first pathway, but then starts overfitting the training data before it can learn non-robust predictive features. At this point, validation accuracy stagnates. The model finishes with a training accuracy of 100% but a validation accuracy of 81%. Nearly all the performance of the model can be attributed to its learning predictive robust features.

**Hyperparameters :** In Section C of the Appendix, we present a study of the effect of different hyperparameters for a Resnet-18 model trained on $\widehat{\mathcal{D}}_{det}$. We observe that the model makes a *sharp* transition from Pathway 1 to 2 in the space of hyperparameters, with a narrow "middle ground".

**On clean data :** Training on the $\widehat{\mathcal{D}}_{det}$ dataset allows us to decompose the accuracy into robust and non-robust, but a similar decomposition doesn't exist for a model trained on $\mathcal{D}$. Instead we utilize Claim 2 and use adversarial transferability as a proxy for whether or not the model has learned non-robust features.

Train two models $\mathcal{M}_1^{(1)}$ and $\mathcal{M}_2^{(1)}$ with different random initializations on the unaltered CIFAR-10 dataset, with both data augmentation and some weight decay. Train two more models $\mathcal{M}_1^{(2)}$ and $\mathcal{M}_2^{(2)}$ with neither weight augmentation nor weight decay. We plot the training and validation accuracies over the course of training for $\mathcal{M}_1^{(1)}$ and $\mathcal{M}_1^{(2)}$ in Figure 4a and Figure 4b.

Simultaneously, we also plot the targeted adversarial attack success, as well as the transfer accuracy to $\mathcal{M}_2^{(1)}$ and $\mathcal{M}_2^{(2)}$. We observe that targeted adversarial attack success is high for both models. However, while adversarial examples generated for $\mathcal{M}_1^{(1)}$ transfer to $\mathcal{M}_2^{(1)}$ with reasonable success, adversarial examples generated for $\mathcal{M}_1^{(2)}$ fail to transfer to either $\mathcal{M}_2^{(1)}$ or $\mathcal{M}_2^{(2)}$.

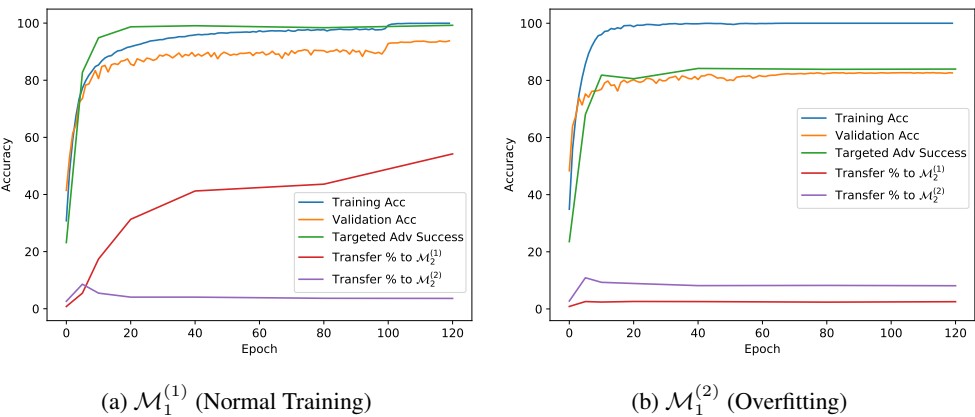

(a) $\mathcal{M}_1^{(1)}$ (Normal Training)        (b) $\mathcal{M}_1^{(2)}$ (Overfitting)

Figure 4: (Best viewed in color). Training and validation accuracy of $\mathcal{M}_1^{(1)}$ and $\mathcal{M}_1^{(2)}$ along with accuracy of targeted adversarial attacks and adversarial transfer accuracy to $\mathcal{M}_2^{(1)}$ and $\mathcal{M}_2^{(2)}$.

We conclude that $\mathcal{M}_1^{(1)}$ learns predictive non-robust features as training progresses, and $\mathcal{M}_2^{(1)}$ does not. Thus, $\mathcal{M}_1^{(1)}$ follows Pathway 1 and $\mathcal{M}_1^{(2)}$ follows Pathway 2. We observe that learning predictive non-robust features seems to be essential for good generalization on the validation set, and robust features alone do not suffice.

**Remark :** Although Figure 1a suggests that the model eventually generalizes *better* to the non-robust feature mapping than the robust, this is not a generally applicable rule over different datasets, architectures and combinations of hyperparameters. Table 5 in the Appendix illustrates this point.

### 4.3 RELATION TO OTHER RESULTS ABOUT NEURAL NETWORK TRAINING

**Low and high learning rates :** The regularizing effect of a high initial learning rate has been studied in detail by Li et al. (2019). They construct a dataset with two types of patterns - those that are *hard-to-fit* but *easy-to-generalize* (i.e., low in variation), and those that are *easy-to-fit* but *hard-to-generalize* (i.e., noisy).

They show that a neural network trained with small learning rate first focuses on the hard-to-fit patterns, and because of the low noise in them, quickly overfits to the training set. As a result, it is not able to learn easy-to-fit pattern effectively later on. In contrast, a model that starts out with a high learning rate learns easy-to-fit pattern first, and since these are noisy, doesn't overfit the training set. Later on, once the learning rate is annealed, the model is able to effectively learn the harder-to-fit patterns.

These two cases can be crudely mapped onto our two pathways. The model in Pathway 2, trained with a low LR, learns only robust features to start out, indicating that these features are hard-to-fit. It overfits the training set and thereafter is unable to learn the non-robust features, which are easy-to-fit.

The model in Pathway 1, trained with a high LR, quickly begins to learn the non-robust features which are easy-to-fit. However, it learns the robust features too alongside, indicating that this mapping from the low and high LR scenarios to our two pathway theory is not perfect.

**Complexity of learned functions :** Another perspective on the training of neural networks is given by Nakkiran et al. (2019). They define a performance correlation metric between two models that captures *how much of the performance of one model can be explained by the other*, and show that as training progresses, the peformance correlation between the current model and the best simple model decreases. This indicates that the functions learned by a neural network become increasingly complex as training progresses.

Although their metric is defined for a binary classification setting, we adapt it for the multi-class setting, and use a multi-class logistic regression classifier as the "best simple classifier". We measure the performance correlation between the model trained on $\mathcal{D}_{det}$ and the simple classifier as training

progresses. The performance correlation, scaled by 100 and smoothed, is shown in Figure 1 along with the training curves.

We observe in Figure 1a that the point at which the performance correlation plateaus corresponds with when the robust accuracy decreases sharply. Similarly in Figure 1b, the correlation levels off along with the robust accuracy. We conjecture that the initial robust features learned by the model are simple, linear functions, and the non-robust features are more complex and non-linear. This is in line with the findings of Tramèr et al. (2017) that transferable adversarial examples occur in a space of high dimensionality.

## 5 THEORETICAL ANALYSIS

In this section, we present some results for gradient descent on a toy linear model on a distribution with robust and non-robust predictive features. We get results that mirror our two pathways, for different choices of initial conditions. The setting we use is an adaptation of the one used by Nakkiran et al. (2019). For proofs of these theorems, refer to Section A of the Appendix.

Define a data distribution $P$ over $\mathbb{R}^d \times \{-1, 1\}$ as follows :

$$y \overset{u.a.r}{\sim} \{-1, 1\}, \quad q \overset{u.a.r}{\sim} \{3, ..., d\}$$

$$\lambda_1 \sim \text{Bernoulli}(p) \text{ over } \{\pm 1\}, \quad \lambda_2 \sim \text{Bernoulli}(p/k) \text{ over } \{\pm 1\}, \quad p \in (0, 1/2), \quad k > 1$$

$$\boldsymbol{x} = y\lambda_1 \boldsymbol{e_1} + \epsilon y \lambda_2 \boldsymbol{e_2} + \boldsymbol{e_q}$$

where $\epsilon < 1$ is some small positive constant, and $\boldsymbol{e_i}$ denotes the $i^{\text{th}}$ natural basis vector. Now sample a "training set" $(X_1, Y_1), ..., (X_n, Y_n)$ from this distribution. Let $A = [X_1^T; ...; X_n^T] \in \mathbb{R}^{n \times d}$ and $B = [Y_1, ..., Y_n]^T \in \mathbb{R}^n$. Consider training a linear classifier using gradient descent with a learning rate of $\eta$ to find $\boldsymbol{w} \in \mathbb{R}^d$ that minimizes the squared loss :

$$\mathcal{L}(\boldsymbol{w}) = \frac{1}{2n} \|B - A\boldsymbol{w}\|_2^2, \quad \boldsymbol{w} \in R^d$$

We operate in the overparameterized setting, where $n \ll d$. So with high probability, the coordinates $\{3, ..., d\}$ of the training data are orthogonal for all the training points. [2]

The idea is that the data consists of a "robust" feature given by the first coordinate, a "non-robust" feature (one that can become anti-correlated with a perturbation of $2\epsilon$) given by the second coordinate, and a noisy component that comprises the rest of the coordinates, making it possible for a model to fit the data exactly. The robust component is predictive of the true label with probability $1 - p$, and the non-robust component is predictive of the true label with probability $1 - (p/k)$. For simplicity, assume that the initial weight vector $\boldsymbol{w}_0 = \boldsymbol{0}$, and that $n$ is sufficiently large.

**Theorem 1** (Robust before Non-robust). *If $\epsilon \leq \sqrt{(1 - 2p)/(1 - 2p/k)}$, then at the end of the first step, with high probability, the model will rely on the robust feature for classification,, i.e., $\boldsymbol{w}_1^{(1)} \geq \epsilon \boldsymbol{w}_1^{(2)}$, and will have a population accuracy of $1 - p$.*

**Theorem 2** (Two Pathways). *Define*

$$k_t = \frac{2p(1 + n\epsilon^2) - 2p(1 - \epsilon)}{2p(1 + n\epsilon^2) - (1 - \epsilon)}$$

*Then if $\eta \leq 2/(1 + \epsilon^2 + (1/n))$, as the number of gradient steps goes to infinity,*

- *if $k \geq k_t$, sample accuracy approaches 1 and population accuracy approaches $1 - (p/k)$ with high probability.*

- *if $k < k_t$, sample accuracy approaches 1 and population accuracy approaches $1 - p$ with high probability.*

*Discussion :* The two cases of Theorem 2 very roughly correspond to Pathways 1 and 2. Since this is a strongly convex problem, gradient descent with a small enough learning rate will converge to a fixed solution, so we cannot mimic the setting where different training hyperparameters lead to Pathway 1 or 2. But we can see that if the non-robust feature is predictive enough, the model learns the non-robust feature, otherwise it learns the robust feature.

---

[2]Caveat : here, $d$ is both the input dimensionality and the number of parameters. Although deep learning models are overparameterized, it is uncommon for datasets to have more dimensions than data points.

| Trained On | On CIFAR-minus-Flickr | On CINIC-10 |
|---|---|---|
| CIFAR-minus-Flickr | 93.4 | 65.7 |
| CINIC-10 | 82.1 | 84.6 |
| CIFAR-minus-Flickr $\mathcal{D}_{det}$ | 33.7 | **20.5** |
| CINIC-10 $\mathcal{D}_{det}$ | **19.6** | 30.1 |

Table 2: Accuracy of Resnet50 models on the CIFAR-minus-Flickr and CINIC-10 test sets. The two numbers in bold are the ones to focus on.

## 6 DIGRESSION : CROSS-DATASET TRANSFER

One view of non-robust features is that they are peculiarities or quirks in the data distribution. We provide evidence that allows us to tentatively refute this assumption by showing that one can construct two datasets from completely independent sources, and a model that learns only the predictive non-robust features of one dataset can achieve non-trivial generalization on the other dataset.

The CINIC-10 dataset Darlow et al. (2018) is a distribution-shifted version of CIFAR-10 constructed by sampling from the ImageNet synsets corresponding to each of the CIFAR-10 classes. Although it may seem like CIFAR-10 and CINIC-10 could be candidates for two datasets drawn from independent sources, ImageNet is constructed by querying Flickr, and Flickr is also one of the seven sources for the 80 million TinyImages dataset (Torralba et al. (2008)) that was used to construct CIFAR-10 (Krizhevsky et al. (2009)). So roughly one in seven CIFAR-10 images is from Flickr.

To be even more certain that there are no spurious correlations creeping in because of a common source, we construct the CIFAR-minus-Flickr dataset that consists of those CIFAR-10 images that haven't been sourced from Flickr. This comprises 52,172 out of the 60,000 CIFAR-10 images.

We construct $\mathcal{D}_{det}$ datasets as described in Section 4 for CIFAR-minus-Flickr and CINIC-10, and train Resnet50 models on them. These models can only learn non-robust features to help them generalize to the original unperturbed datasets, because the robust features are correlated with the *shifted* labels.

The results are shown in Table 2. Both $\mathcal{D}_{det}$ trained models achieve an accuracy of close to 20% on the other dataset, which is a long way from the expected 10% accuracy of a random model.

## 7 CONCLUSION AND FUTURE DIRECTIONS

In this paper, we've shown that from the perspective of predictive robust and non-robust features, neural network training follows two very different pathways, corresponding to the *training* and *overfitting* regimes. In both regimes, the model starts out by learning predictive robust features first.

This decomposition into two distinct pathways has several interesting implications. For instance, adversarial transferability means that even an adversary with no access to a model can mount a successful attack by constructing adversarial examples for a proxy model. But a model trained via Pathway 2 learns no predictive non-robust features, and adversarial examples generated for another model will in general not transfer to this model. Thus an adversary cannot perform a successful attack on this model without at least the ability to query the model and observe its outputs for a large number of inputs.

A line of enquiry that arises naturally from our work is understanding precisely why this behavior occurs in neural networks. What characterstics do predictive non-robust features have that ensure that they are learned only subsequent to predictive robust features? We pose finding a more precise definition of non-robust features that will allow us to theoretically analyze and explain these properties as an important direction for future work.

Finally, as we show in Section 6, predictive non-robust features can occur across datasets sampled from independent sources. Although this needs to be investigated more thoroughly, our results challenge the view that non-robust features are peculiarities of the data distribution. We speculate that some of these features could have a meaning in terms of human semantics, like our illustrative example where the eye color was a predictive non-robust feature.

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

## A PROOFS OF THEOREMS IN SECTION 5

Consider using Gradient Descent with a learning rate of $\eta$ to minimize the squared loss as described in Section 5. Then,

$$\boldsymbol{w}_{t+1} := \boldsymbol{w}_t + \frac{1}{n}\eta A^T(B - A\boldsymbol{w}_t)$$

Letting $\alpha = \eta/n$, it can be proved by induction that

$$\boldsymbol{w}_t = \boldsymbol{w}_0 + \alpha A^T \left[\sum_{k=0}^{t-1}(I - \alpha AA^T)^k\right](B - A\boldsymbol{w}_0)$$

Let the largest eigenvalue of $AA^T$ be $\lambda_{max}$. If $|1 - \alpha\lambda_{max}| \leq 1$, then

$$w_t = w_0 + A^T(AA^T)^{-1}\left[I - (I - \alpha AA^T)^t\right](B - Aw_0)$$

$$\implies \boldsymbol{w}_T = A^T(AA^T)^{-1}(B - A\boldsymbol{w}_0) + \boldsymbol{w}_0 \tag{10}$$

for some very large number of steps $T$. This achieves zero empirical training error, as we can verify that $A\boldsymbol{w}_T = B$.

Let the first and second column of $A$ be $s$ and $r$ respectively. Since (with high probability) coordinates $3$ to $d$ are orthogonal for all training points,

$$AA^T = I + ss^T + rr^T$$

Other than 1, the eigenvalues of this matrix are

$$1 + \frac{(s^T s + r^T r) \pm \sqrt{(s^T s - r^T r)^2 + 4(s^T r)^2}}{2}$$

It's easy to see that $s^T s = n$, $r^T r = n\epsilon^2$. Let $s^T r = r^T s = n\epsilon\beta$. Then since $\epsilon$ is small,

$$\sqrt{(s^T s - r^T r)^2 + 4(s^T r)^2} = n\sqrt{(1 - \epsilon^2)^2 + 4(\epsilon\beta)^2}$$
$$\approx n(1 + \epsilon^2(2\beta^2 - 1))$$

Then the eigenvalues are

$$\lambda_1 = 1 + n(1 + \epsilon^2\beta^2), \quad \lambda_2 = 1 + n\epsilon^2(1 + \beta^2)$$

**Theorem 1 :** *If $\epsilon \leq \sqrt{(1 - 2p)/(1 - 2p/k)}$, then at the end of the first step, with high probability, the model will rely on the robust feature for classification,, i.e., $w_1^{(1)} \geq \epsilon w_1^{(2)}$, and will have a population accuracy of $1 - p$.*

*Proof.*

$$w_1^{(1)} = w_0^{(1)} + \alpha s^T (B - A w_0)$$
$$w_1^{(2)} = w_0^{(2)} + \alpha r^T (B - A w_0)$$

$$w_1^{(1)} \geq \epsilon w_1^{(2)} \implies \alpha s^T B \geq \epsilon \alpha r^T B$$

It is easy to see that

$$\mathbb{E}\left[\frac{1}{n} s^T B\right] = (1 - 2p), \quad \mathbb{E}\left[\frac{1}{n} \epsilon r^T B\right] = \epsilon^2 \left(1 - \frac{2p}{k}\right)$$

Since $n$ is sufficiently large, these random variables are close to their mean with high probability. So,

$$(1 - 2p) \geq \epsilon^2 \left(1 - \frac{2p}{k}\right)$$

This is true by the assumed bound on $\epsilon$. $\qquad\square$

**Theorem 2 :** *Define*
$$k_t = \frac{2p(1 + n\epsilon^2) - 2p(1 - \epsilon)}{2p(1 + n\epsilon^2) - (1 - \epsilon)}$$
*Then if $\eta \leq 2/(1 + \epsilon^2 + (1/n))$, as the number of gradient steps goes to infinity,*

- *if $k \geq k_t$, sample accuracy approaches $1$ and population accuracy approaches $1 - (p/k)$ with high probability.*
- *if $k < k_t$, sample accuracy approaches $1$ and population accuracy approaches $1 - p$ with high probability.*

*Proof.* Gradient descent will converge if

$$\alpha\lambda_1 \leq 2 \implies \eta \leq \frac{2}{1 + \epsilon^2\beta^2 + (1/n)}$$

Using the fact that $\beta \leq 1$ gives us the bound on learning rate in the theorem statement.

Next, using Equation 10,

$$w_T = A^T (I + ss^T + rr^T)^{-1} (B - A w_0) + w_0$$

$$(I + ss^T + rr^T)^{-1} = \left( I - \frac{(1 + s^T s)(rr^T) + (1 + r^T r)(ss^T) - (s^T r)(sr^T) - (r^T s)(rs^T)}{(1 + s^T s)(1 + r^T r) - (s^T r)(r^T s)} \right)$$

$$\implies w_T^{(1)} = s^T (I + ss^T + rr^T)^{-1}(B - Aw_0) + w_0^{(1)}$$
$$= \left[ s^T - \frac{(1 + n)(n\epsilon\beta)r^T + n(1 + n\epsilon^2)s^T - (n^2\epsilon\beta)r^T - (n^2\epsilon^2\beta^2)s^T}{(1 + n)(1 + n\epsilon^2) - n^2\epsilon^2\beta^2} \right] B$$
$$= \left[ \frac{(1 + n\epsilon^2)s^T - (n\epsilon\beta)r^T}{(1 + n)(1 + n\epsilon^2) - n^2\epsilon^2\beta^2} \right] B$$

$$w_T^{(2)} = r^T (I + ss^T + rr^T)^{-1}(B - Aw_0) + w_0^{(2)}$$
$$= \left[ r^T - \frac{(1 + n)(n\epsilon^2)r^T + (1 + n\epsilon^2)(n\epsilon\beta)s^T - (n^2\epsilon^2\beta^2)r^T - (n^2\epsilon^3\beta)s^T}{(1 + n)(1 + n\epsilon^2) - n^2\epsilon^2\beta^2} \right] B$$
$$= \left[ \frac{(1 + n)r^T - (n\epsilon\beta)s^T}{(1 + n)(1 + n\epsilon^2) - n^2\epsilon^2\beta^2} \right] B$$

where we have used the fact that $w_0 = 0$. Now suppose we sample a new point $(X, Y)$ from the data distribution. Let $q_i$ denote the index of the noise coordinate of $X_i$ and let $q$ denote the index of the noise coordinate of $X$. With high probability, $q \neq q_i \; \forall i$. So,

$$X^T w_T = X^{(1)} w_T^{(1)} + X^{(2)} w_T^{(2)} + X^{(q)} w_T^{(q)}$$
$$= X^{(1)} w_T^{(1)} + X^{(2)} w_T^{(2)} + w_0^{(q)}$$
$$= X^{(1)} w_T^{(1)} + X^{(2)} w_T^{(2)}$$

We want to analyze the case when the first and second coordinate *disagree*. Let $X^{(1)} = -1$ and $X^{(2)} = \epsilon$. In this scenario if the model always predicts $X^T w_T \geq 0$, it will match the prediction of the *second* coordinate and achieve a population accuracy of $1 - p/k$. On the other hand if it always predicts $X^T w_T < 0$, it will match the prediction of the *first* coordinate and achieve a population accuracy of $1 - p$.

$$X^T w_T > 0 \implies \epsilon w_T^{(2)} > w_T^{(1)}$$

$$\implies \epsilon \left[ \frac{(1 + n)r^T - (n\epsilon\beta)s^T}{(1 + n)(1 + n\epsilon^2) - n^2\epsilon^2\beta^2} \right] B \geq \left[ \frac{(1 + n\epsilon^2)s^T - (n\epsilon\beta)r^T}{(1 + n)(1 + n\epsilon^2) - n^2\epsilon^2\beta^2} \right] B$$

With high probability, $s^T B = n(1 - 2p)$, $r^T B = n\epsilon(1 - 2p/k)$, and $\beta = (1 - p(k+1)/k + 4p^2/k)$.

$$\implies \epsilon \left[ (1 + n)n\epsilon \left( 1 - \frac{2p}{k} \right) - (n\epsilon\beta)n(1 - 2p) \right] \geq \left[ (1 + n\epsilon^2)n(1 - 2p) - (n\epsilon\beta)n\epsilon \left( 1 - \frac{2p}{k} \right) \right]$$

$$\implies n(\epsilon^2 - 1) + 2pn \left( 1 - \frac{\epsilon^2}{k} \right) + 2pn^2\epsilon^2 \left( 1 - \frac{1}{k} \right) \geq 0$$

$$\implies k \geq \frac{2p(1 + n\epsilon^2) - 2p(1 - \epsilon)}{2p(1 + n\epsilon^2) - (1 - \epsilon)}$$

$\square$

# B    OTHER DATASETS AND ARCHITECTURES

In this section, we provide training graphs illustrating Pathway 1 and 2 for **Resnet18** and **Resnet50** models trained on the $\mathcal{D}_{det}$ versions of the **CIFAR-10** and **CINIC-10** (Darlow et al. (2018)) datasets. Along with each graph, we mention the hyperparameters used.

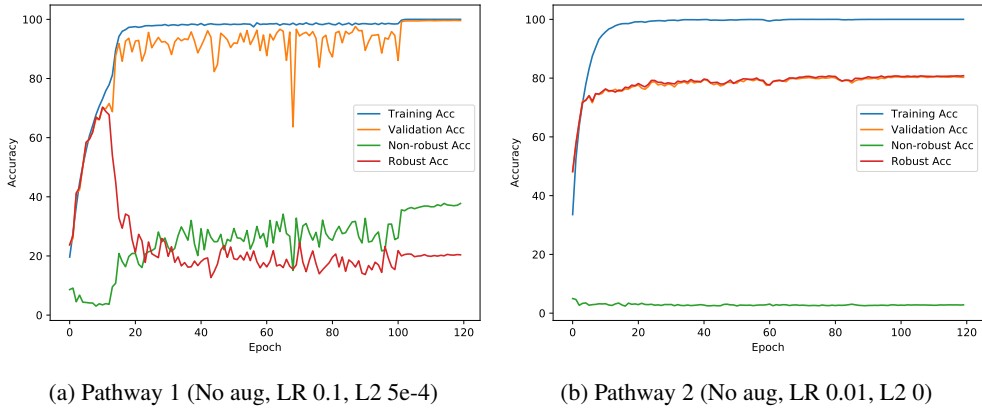

(a) Pathway 1 (No aug, LR 0.1, L2 5e-4)    (b) Pathway 2 (No aug, LR 0.01, L2 0)

Figure 5: Resnet50, CIFAR-10 $\mathcal{D}_{det}$

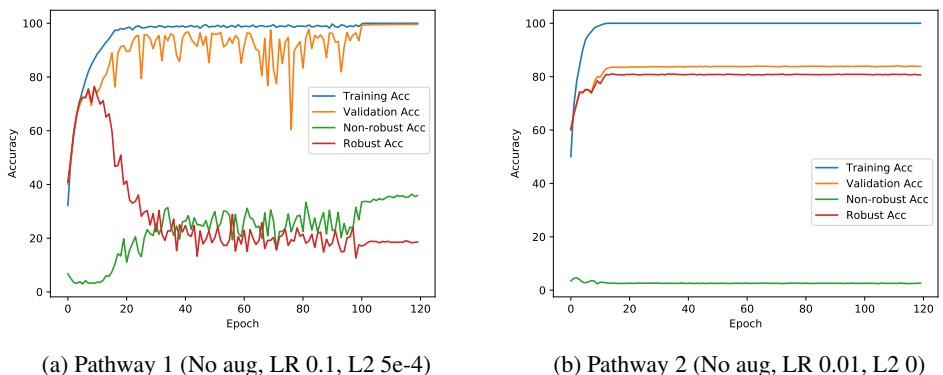

(a) Pathway 1 (No aug, LR 0.1, L2 5e-4)    (b) Pathway 2 (No aug, LR 0.01, L2 0)

Figure 6: Resnet18, CIFAR-10 $\mathcal{D}_{det}$

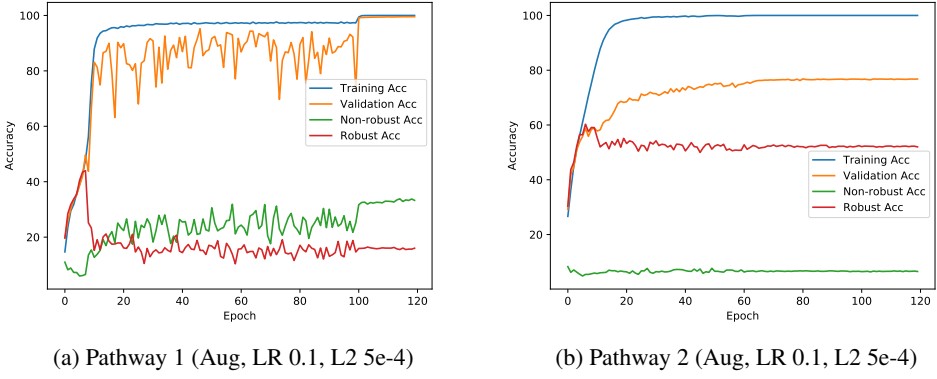

(a) Pathway 1 (Aug, LR 0.1, L2 5e-4)    (b) Pathway 2 (Aug, LR 0.1, L2 5e-4)

Figure 7: Resnet50, CINIC-10 $\mathcal{D}_{det}$

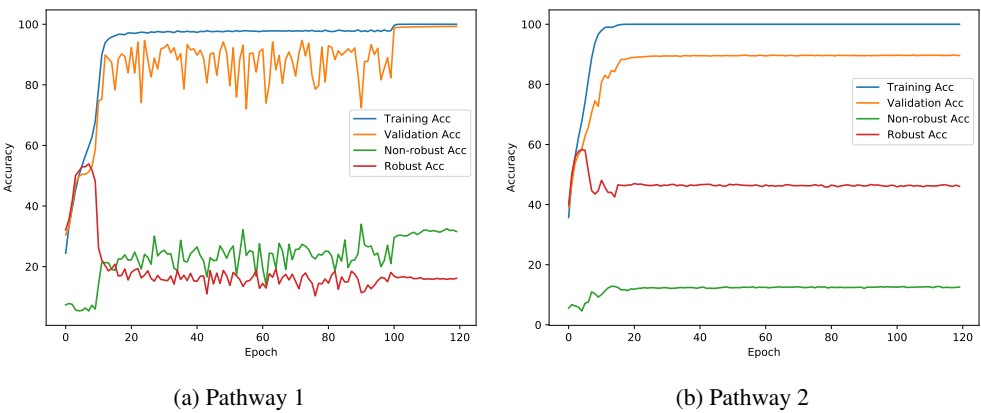

(a) Pathway 1                 (b) Pathway 2

Figure 8: Resnet18, CINIC-10 $\mathcal{D}_{det}$

## C    HYPERPARAMETERS

In this section, we list the hyperparameters used in our experiments. We also take the case of a Resnet18 trained on $\mathcal{D}_{det}$ CIFAR-10 and look at which hyperparameters lead to Pathway 1 and which lead to Pathway 2. As we note in Section 4, there is a *sharp* transition between the two pathways in the space of hyperparameters.

**Common Parameters :**

| Parameter | Value |
|-----------|-------|
| Batch Size | 128 |
| LR Annealing | $\times 0.1$ at Epoch 100 |
| Optimizer | SGD |
| Momentum | 0.9 |
| PGD norm | $L_2$ |
| PGD epsilon | 0.5 |
| PGD steps | 100 |
| Step size | 0.1 |

Table 3: These parameters were used throughout all our experiments

**Training on Non-transferable examples (Section 3)**

We trained the model using 120 epochs of SGD, and did a grid search over the following combinations of hyperparameters.

| Parameter | Values |
|-----------|--------|
| Learning Rate | {0.01, 0.1} |
| Data Augmentation | {True, False} |
| L2 Weight Decay | {0, 1e-5, 5e-4} |

Table 4: Hyperparameter grid for training on non-transferable adversarial examples

**CIFAR $\mathcal{D}_{det}$ (Figure 1)** :

- Pathway 1 : LR 0.1, No data augmentation, L2 5e-4

- Pathway 2 : LR 0.01, No data augmentation, L2 0

**CIFAR (Figure 4)** :

- Pathway 1 : LR 0.01, Data augmentation, L2 5e-4

- Pathway 2 : LR 0.01, No data augmentation, L2 0

**Which hyperparameters lead to Pathway 1 and Pathway 2? :**

We pick one model (Resnet18) and one dataset (CIFAR-10), and explore which hyperparameters lead to Pathway 1 and which lead to Pathway 2. To re-iterate, Pathway 2 is characterized by the model learning almost exclusively only the robust features. All models were trained with SGD for 120 epochs, and the reported accuracies are after the final epoch.

| Data Aug | L2 decay | Learning Rate | |
|---|---|---|---|
| | | 0.1 | 0.01 |
| Yes | 0 | 30.9 / **43.0** | 24.4 / **50.66** |
| | 5e-4 | **38.3** / 26.2 | 30.9 / **43.0** |
| No | 0 | 2.9 / **77.8** | 2.6 / **80.7** |
| | 1e-5 | 4.2 / **79.1** | – |
| | 5e-5 | 2.7 / **80.1** | – |
| | 1e-4 | 2.6 / **79.7** | – |
| | 2e-4 | 4.0 / **73.7** | – |
| | 2.25e-4 | 18.6 / **43.4** | – |
| | 2.5e-4 | 13.5 / **49.3** | – |
| | 2.75e-4 | **30.4** / 24.7 | – |
| | 3e-4 | 27.1 / **30.6** | – |
| | 5e-4 | **35.9** / 18.6 | 20.9 / **43.7** |

Table 5: (Test / Shifted Test) accuracy for a Resnet18 model trained on CIFAR-10 $\mathcal{D}_{det}$. Corresponds to (Non-robust Accuracy / Robust Accuracy). The higher of the two is **bold**, and the configurations which follow Pathway 2 have been highlighted in green .

*Remark :* As we can see in the case with LR 0.1 and no data augmentation, the network exhibits a *sharp* transition from Pathway 2 to Pathway 1 in the space of hyperparameters. There is a narrow "middle ground" around L2 = 2.5e-4.

**Cross-Dataset Transfer (Table 2)** :

- CIFAR-minus-Flickr (clean) : *Adam optimizer*, LR 1e-3, L2 1e-5, with data augmentation.
- CINIC-10 (clean) : LR 0.01, data augmentation, L2 5e-4.
- CIFAR-minus-Flickr $\mathcal{D}_{det}$ : LR 0.1, No data augmentation, L2 5e-4.
- CINIC-10 $\mathcal{D}_{det}$ : LR 0.1, No data augmentation, L2 5e-4.

