# OpenReview forum: "SGD on Neural Networks learns Robust Features before Non-Robust"
_ICLR.cc/2021/Conference — Reject_

### Official Review · AnonReviewer2 · 2020-10-26
**Interesting findings, but I feel it questionable**

**Rating:** 5
**Confidence:** 4

**Review:**

This paper reveals interesting phenomenons of the training dynamics of SGD on convolutional neural networks for image classification. From carefully designed experiments and simplified theories, it argues that SGD in this setting follows two different pathways, depending on the regularizations and hyperparameters. In the first pathway where regularizations are applied and initial learning rate is large, the network first learns predictive robust features and weakly-predictive non-robust features, and then learns to incorporate predictive non-robust features. In the second pathway where little or no regularization is applied and initial learning rate is small, the network does the same thing as the first pathway in the beginning, but then starts overfitting to noise. Overall, I think this paper may have discovered something interesting, but I also feel uncertain about the correctness of the messages due to the lack of clearness on the experimental results. I strongly recommend the authors to improve the clearness in the next version.

Strengths:
1. The phenomenon demonstrated in this paper seems to be interesting. The authors found that CNNs with SGD behave differently in using robust and non-robust features under different hyperparameters.

2. Simplified theoretical analysis is provided to justify the findings. But I have not carefully checked the correctness.

Weaknesses:
1.  My major concern is about the clearness of the experimental results, from which I cannot tell whether the conclusions are correct. For all the plots, there is not any written definition for the metrics such as "Shifted Test Acc" and "Transfer % to Path 1", and it is quite difficult to guess the definitions given the complicated setups, which makes me confused and feel quite uncertain about my understanding of the paper. For example, in Figure 2, I can guess what "shift" means, but I am not sure whether these accuracies are evaluated on the clean samples or the adversarially perturbed samples from a certain network, and on which subset are the networks tested. With Figure 2, the authors claim that "the network trained on the examples that transfer generalizes well to the true distribution". If the accuracies are evaluated on clean samples, then I cannot understand why "Test Acc" is higher than "Shifted Test Acc" when the model is trained with "All Examples (66%)". Intuitively, even the training examples are perturbed, the model should have higher "Shifted Test Acc" than "Test Acc", since the adversarial examples should have more robust features for the original class than the shifted class. If it is evaluated on perturbed samples, then the network might have just learned to depend on the adversarial perturbations, since we already know the perturbations for "Non-Transferable (33%)" do not transfer well but those for "All Examples (66%)" transfer well. If Figure 2 cannot justify "a neural network being unable to learn a generalizing model as evidence that there are no highly predictive non-robust features", then all the following experimental results will be questionable.

2. The definition of robust and non-robust features in Section 2 are confusing, not related to the experimental and theoretical results, and might have some minor issues. First, from $\mathbb{E}[y\cdot f_{NR}(x)]>0$ and $\mathbb{E}[\inf_{\delta\in \Delta(x)} y\cdot f_{NR}(x+\delta)]\ll 0$, we cannot conclude that $\mathbb{E}[\inf_{\delta\in \Delta(x)} y\cdot f_{NR}(x+\delta)]\gg 0$, since we know for $\delta=0$,  $\mathbb{E}[-y\cdot f_{NR}(x+0)]<0$. Second, the sum $f_C$ of the robust $f_R$ and weakly-predictive non-robust $f_{NR}$ does not necessarily satisfy $\mathbb{E}[\inf_{\delta\in \Delta(x)} y\cdot f_{C}(x+\delta)] < 0$. Let $\delta_1=\arg\min_{\delta\in \Delta(x)} y\cdot f_{R}(x+\delta)$, $\delta_2=\arg\min_{\delta\in \Delta(x)} y\cdot f_{NR}(x+\delta)$. Given the conditions, it is possible that $f_{NR}(x+\delta_1)>0$, and $f_R (x+\delta_2)\gg 0$, and therefore it is possible that $\mathbb{E}[\inf_{\delta\in \Delta(x)} y\cdot f_{C}(x+\delta)] > 0$, i.e., the "contamination" does not always happen when robust and weakly-predictive non-robust features are both present, which might also be consistent with Remark 1. Section 2 is too confusing that I do not quite follow the motivation for Definition 1, and cannot find where Definition 1 is used later in the paper.

3. Figure 1 (b) seems to conflict with Figure 3 (b). Figure 1 (b) shows the robust accuracy is around 80% at convergence, but Figure 3 (b) says the targeted adv success rate is around 80%, meaning the robust accuracy should be around 20%.

4. The number of hyperparameters tried are not dense enough to show "there is no middle ground". I would suggest testing at least 20 points within the current interval for each hyperparameter including lr and weight decay.

5. The title might be a bit misleading. It only describes the process of pathway 1, not pathway 2.

---
### Updates after rebuttal
I appreciate the authors' efforts in revising the paper into a more rigorous and illustrative way. However, I still feel the current version difficult to follow, and I am not sure whether the conclusions are really correct or not. The conclusions are drawn from assumptions for empirical observations, but the scale of the experiments is not large enough, and the conditions for the assumptions are not specified rigorously. Therefore, I feel quite reluctant in raising my scores.

My final suggestions for improvements in the future version:

1. Instead of using accuracies on $\mathcal{D}_{shift}$ as a proxy for how much of the model’s performance can be attributed to its learning predictive robust features, showing the accuracy of the model on the shifted dataset under adversarial attacks would make it more convincing to me.

2. I would prefer the figures to be close to the text that elaborate the figures. For example, Figure 1 is in Page 2, but the details of this complicated figure is not discussed until Sec. 4.2.

---

> ### Author Response · Authors · 2020-11-20
> **Response to AnonReviewer2**
>
> Thanks for the review! Here are our responses to the concerns you raised.
>
> **On the clarity of experimental results :**
>
> We have made extensive changes to Sections 2, 3 and 4 which we believe will make the paper more precise and easier to understand. We provide a summary of these changes in a reply to the main thread.
>
> **On Figure 2 :**
>
> The evaluations on Figure 2 are indeed on clean samples. The “shifted” test accuracy is a measure of how well the model has learned the robust features of the original dataset, and the test accuracy is a measure of how well the model has learned the non-robust features. If a model has higher Test Accuracy than Shifted Test Accuracy, then this means that it has learned to utilize the non-robust features in preference to the robust. However, this is not a generally applicable rule across datasets and models.
>
> We hope that the additional discussion in Section 3 as well as the illustrative example will help clarify why this behavior occurs.
>
> **On the definitions in Section 2 :**
>
> Thank you for pointing out an issue with $\mathbb{E} [\inf_{\delta \in \Delta(x)} -y \cdot f_{NR}(x+\delta) ] \gg 0$ ! The “inf’’ should have been “sup” because of the negative sign. That equation has now been removed from the paper after revisions because of space constraints.
>
> However, there is no issue with the next set of equations. We are concerned with the infimum over $\delta$ when it comes to $f_C$. Using your notation, if $f_{NR}(x + \delta_2) \ll 0$ and $f_R(x + \delta_2) > 0$, then $f_C(x + \delta_2) < 0$ for this value of $x$. It is thus not relevant that $f_{NR}(x + \delta_1) > 0$.
>
> Regarding Definition 1, please refer to the reply to the main thread.
>
> **On robust accuracy in Figures 1 and 3 :**
>
> We realized that the term “robust accuracy” is potentially confusing because it suggests that it means “Accuracy under adversarial attack”, which is not true. We have therefore replaced it with “robust feature accuracy”, indicating "accuracy derived from robust features", and likewise for the term "non-robust accuracy".
>
> Targeted adversarial attack success could be 100% even if the model learns only predictive robust features, because these can have non-robust contaminants that open up vulnerabilities in the model.
>
> Finally, Figures 1 and 3 depict models trained on different data distributions. The Figure 1 model is trained on relabeled adversarial examples, while the Figure 3 model is trained on the original CIFAR-10 dataset.
>
> **On whether there is a middle ground :**
>
> Thank you for the suggestion. We are currently running experiments with more hyperparameters, and we will update the paper with the results.
>
> **On the choice of title :**
>
> In both Pathway 1 and Pathway 2, the model starts off by learning predictive robust features in preference to predictive non-robust. This is what our title alludes to.

---

> > ### Comment · AnonReviewer2 · 2020-11-24
> > **Further questions**
> >
> > Thanks for the feedback. While I have not been able to read all the details, I still hold the opinion that $f_C$ is not necessarily non-robust, given the definitions of $f_C, f_{NR},f_{R}$ on page 3 (please add numbers to these equations unless you feel they are not relevant, thanks!). $f_C$ could still be robust when $f_R$ gives very confident predictions at points where $f_{NR}$ is the most vulnerable. Specifically, suppose $\delta$ is the minimizer for $y\cdot f_C$ at $x$. Given these definitions, it is possible that for every $x$, $y\cdot f_{NR}(x+\delta)<0$, but $y\cdot f_{R}(x+\delta)>0$, and $y\cdot [f_{NR}(x+\delta)+f_{R}]>0$. In this way, $f_C$ is still robust.
> >
> > Since the following analysis depends on this contamination argument, I think it is important to make sure it is correct in the first place.

---

> > > ### Author Response · Authors · 2020-11-24
> > > **There is a minor loophole, thank you for noticing it!**
> > >
> > > You are right, there is a minor loophole in that set of equations. Thank you for noticing it! However, while it's not *necessary* that $f_C$ is non-robust, it is definitely *possible* to construct a scenario where $f_C$ is non-robust. The purpose of our argument in Equations 6, 7 and 8 is simply to outline a possible scenario in which contaminated robust features can be formed.
> > >
> > > Equations 6 and 7 allow us to conclude that
> > > $$\mathbb{E}\left[ \inf_{\delta \in \Delta(x)} y \cdot f_{NR}(x + \delta) + \inf_{\delta \in \Delta(x)} y \cdot f_{R}(x + \delta)\right] < 0$$
> > >
> > > If $\delta_1(x)$ achieves $\inf_{\delta \in \Delta(x)} y \cdot f_{NR}(x + \delta)$ for each $x$, we would like to say that
> > > $$\mathbb{E}\left[ y \cdot f_{NR}(x + \delta_1(x)) + y \cdot f_{R}(x + \delta_1(x))\right] = \mathbb{E}[ y \cdot f_C (x + \delta_1(x))] < 0$$
> > >
> > > This is true if $f_{R}(x + \delta_1(x))$ is not too large. As you put it, we would like it if $f_R$ did not give very confident predictions at points where $f_{NR}$ is the most vulnerable. We have amended the paragraph on Page 3 to say "Then it is possible to construct a scenario in which...", instead of stating it as an absolute truth.
> > >
> > > We have also added equation numbers to all relevant equations.

---

> > ### Author Response · Authors · 2020-11-25
> > **Middle ground**
> >
> > We've updated Section C of the appendix with the results of training with more hyperparameters. We observe that there does in fact seem to be a narrow middle ground around L2 = 2.5e-4, where the model's behavior is hard to classify as Pathway 1 or Pathway 2. We've also updated the discussion in Section 4 to reflect this.

---

### Official Review · AnonReviewer3 · 2020-10-27
**Review - Interesting topic and experiments, paper could benefit from improved clarity and organization**

**Rating:** 5
**Confidence:** 4

**Review:**

Summary:

This work studies the learning dynamics of neural networks in terms of robust and non-robust features. In particular, the authors argue that depending on various factors (e.g. learning rate, data augmentation), neural networks will have learning dynamics according to 1 of 2 pathways. Neural networks will either (1) first learn predictive robust features and weakly predictive non-robust features, followed by predictive non-robust features; or (2) only learn robust features, then overfit the training set, thereby not learning non-robust features. The paper has a good discussion expanding upon the robust/non-robust features model of Ilyas et. al. 2019, interesting experiments measuring what features the models is learning, and a digression that presents further results on non-robust features in different datasets.

Recommendation and Explanation:

I would currently recommend a rejection, but I believe the paper is on the borderline and may be improved after clarifications.

The strengths of this paper to me are the discussion in section 2 regarding the robust and non-robust features model of Ilyas et. al, and the experiments measuring “robust accuracy” (accuracy derived from robust features only) and “Non-robust Accuracy” (accuracy derived from non-robust features only). It was interesting to see the discussion of contaminated robust features and the distinction between weakly predictive and highly predictive non-robust features. The experiment relating adversarial transferability and non-robust features is clear and shows that transferable adversarial examples likely have more non-robust features than non-transferable adversarial examples. Overall, the paper has good ideas and studies a topic that merits further investigation.

The weaknesses of this paper to me are unclear explanations of Li et. al 2019, some confusion I have about Pathway 1 vs. Pathway 2, and problems with the organization of the paper that could be improved (e.g. a definition that was abandoned after its introduction). After reading the paper, I have many more questions regarding what the authors did. In rough order of their appearance in the paper:

Section 2: In Ilyas et. al, don’t their experiments show that models generalize to the original distribution (and not to the distribution with flipped labels)? Does this mean that there are not many “contaminants” being learned? How can you evaluate if a robust feature contains “contaminants”?

Section 2: You propose a new definition to exclude contaminated robust features (which are non-robust features), but I did not see you use it again. Is there a way to leverage this definition to further our understanding?

Section 4.1: A diagram explaining what each type of accuracy means could improve clarity.

Section 4.2: How many of the changes described (LR, weight decay, data augmentation) are necessary for transitioning from Pathway 1 to Pathway 2? Can you perform an ablation study to see if any one factor is the most important?

Section 4.2: Is Pathway 2 ever preferable to Pathway 1? Or does robust accuracy matter less than overall accuracy? Explaining when one might be better than the other would be insightful.

Section 4.3: I feel that using the terminology “easy-to-generalize” (as is written in the Li et. al 2019 paper) as opposed to “well-generalizing” is more clear. “Well-generalizing”/”not so well-generalizing” seems to imply that those features will lead to good/bad generalization accuracy, but the Li et. al 2019 paper describes “easy-to-generalize”/”hard-to-generalize” features as those that are not noisy/noisy, and thus easy/hard to learn from and generalize from. In Li et. al 2019, my understanding is that you can get good generalization accuracy from both easy-to-generalize and hard-to-generalize features; one is simply easier to learn from. Furthermore, Li et. al does not claim that all hard-to-fit features are easy-to-generalize or vice versa; they simply analyze how learning rate matters when the dataset does satisfy those properties. Thus, the authors of this paper need to significantly improve their discussion of Li et. al 2019 and how their work relates to it.

Section 5: Isn’t it unlikely that n<<d in deep learning? If so, this should be mentioned as a caveat.

Section 6: Explaining why the D_rand experiment was not done for earlier sections should be placed earlier, rather than in Section 6.

If these questions can be clarified, I believe that will make the paper better.


Post Rebuttal Update:

After reading the author's rebuttal as well as discussion with other reviewers, I will maintain my score. I do appreciate the changes the authors made, and believe that they improve the paper (especially the new example with cat and dog features and the associated figure). I would encourage the authors to incorporate the remaining feedback, as it will be helpful as well.

---

> ### Author Response · Authors · 2020-11-20
> **Response to AnonReviewer3**
>
> Thanks for the review! Here are our responses to the concerns you raised.
>
> **On why Ilyas et al. don’t detect contaminants :**
>
> This is an excellent observation! Ilyas et al. construct adversarial examples for a network (call this N1) using PGD, and train a network (call this N2) on relabeled adversarial examples. N2 then achieves non-trivial generalization to the original distribution. But there are three things that they did that are crucial to their observations :
>
> 1. N1 must have learned predictive non-robust features (must have been trained via Pathway 1).
> 2. Adversarial examples must be constructed by solving the optimization problem in Equation 3 of our paper (using standard PGD, for instance).
> 3. N2 must be trained with some appropriate regularization (must be trained via Pathway 1).
>
> In general, the litmus test to differentiate true non-robust features from contaminants is to check whether N2 generalizes to the true distribution or to the distribution with flipped/shifted labels.
>
> Nakkiran (2019) showed that if condition (2) is violated, i.e., if you run PGD on N1 with a transfer penalty, then N2 generalizes to the distribution with shifted labels and **not** the true distribution. The adversarial examples that this procedure finds thus exploit contaminated robust features. We additionally show that even standard PGD find some examples with identical properties to Nakkiran (2019).
>
> If we are presented with a network N1 and are asked to check whether N1 has learned contaminated robust features or true non-robust features, we would :
>
> 1. Construct adversarial examples for N1 using standard PGD
> 2. Train a network N2 on relabeled adversarial examples, using appropriate regularization and cross-validation to avoid finite sample overfitting.
>
> If N2 generalizes to the true distribution to a non-trivial extent, then N1 has learned predictive non-robust features. If N2 fails to do so and instead generalizes to the distribution with flipped labels, then N1 has learned contaminants.
>
> **On Definition 1 :**
> Please see the reply to the main thread.
>
> **On a diagram in Section 4.1 :**
> A diagram has been added.
>
> **On the effect of different hyperparameters :**
>
> Table 5 in the Appendix studies the effect of different hyperparameters on the choice of Pathway 1 or Pathway 2 for a Resnet18 model trained on CIFAR-10 $\mathcal{D}_{det}$. However, the trends observed in this table are not in general valid across datasets and models. For example, it would seem from Table 5 that data augmentation is sufficient to achieve Pathway 1. But in Figure 7(b), we see that despite data augmentation, the Resnet 50 model trained on CINIC goes down Pathway 2. In particular, there did not seem to be any well-defined rules on the choice of hyperparameters that could determine a priori whether the model would follow Pathway 1 or Pathway 2.
>
> **On when Pathway 2 is preferable to Pathway 1 :**
>
> In the Conclusion, we have added a practical case when we might be interested in training a model via Pathway 2.
>
> **On the discussion of Li et al (2019) :**
>
> Thank you for pointing this out! “Easy-to-generalize” indeed does not mean “well-generalizing”. We have amended our discussion to reflect this, as well as the fact that not all features are one of these two types.
>
> **On $n \ll d$ :**
>
> Deep learning models generally work in the overparameterized setting, so this is actually realistic. For instance, Resnet50 contains 23 million parameters, and CIFAR-10 contains 50,000 images.
>
> **On Drand :**
> Thanks for the suggestion; we have moved the $\mathcal{D}_{rand}$ paragraph into Section 2.

---

> > ### Comment · AnonReviewer3 · 2020-11-20
> > **Thank you for the reply and the changes. I have a few small clarifying questions.**
> >
> > For discussion of Li et al 2019, I noticed that the introduction still uses the terminology "well generalizing" - I would recommend amending this as well.
> >
> > On  n << d:
> >
> > "Deep learning models generally work in the overparameterized setting, so this is actually realistic. For instance, Resnet50 contains 23 million parameters, and CIFAR-10 contains 50,000 images."
> >
> > When reading that section, I understood "d" as referring to the dimensionality of the training datapoints. So for the CIFAR-10 example, wouldn't d = 3x32x32 ~ 3072, and n = 50000? Please let me know if I am misunderstanding something.

---

> > > ### Author Response · Authors · 2020-11-20
> > > **"Well-generalizing", and n << d**
> > >
> > > Thank you for noticing the "well-generalizing" term in the introduction. We have modified it too.
> > >
> > > Regarding $n \ll d$, we are actually using a linear model for classification where $d$ is both the dimensionality of the input as well as the number of parameters of the model. But we realized that the section as it had been written described the model only *after* stating that $n \ll d$, which made it difficult for a reader to understand why this was an over-parameterized setting.
> > >
> > > We have now changed the order of paragraphs in Section 5 such that the model is described *before* mentioning the $n \ll d$ condition. Additionally, we have added a footnote to the effect that it is uncommon for datasets to have more dimensions than data points. Thanks for bringing that to our attention!

---

### Official Review · AnonReviewer4 · 2020-10-28
**hard to interpret the contribution of this paper**

**Rating:** 4
**Confidence:** 4

**Review:**

The paper posits some phenomena on neural network training: 1. With some proper regularizing effect, NN training tends to learn predictive robust features (and weakly predictive non-robust features) first and non-robust features next. 2. Without regularization, NN training does a similar thing as case 1 first but does not learn predictive non-robust features and overfits the training examples.


I find the results vague and hard to interpret. The paper is written in a sloppy way. The intuition is clear but many experimental settings are quite hard to follow. Specifically, the paper constantly omits names of the model or dataset mentioned in the context and refer to them with "it" or very long descriptive language later, which is quite ambiguous or informal. The discussions and results are not very well-organized. It is also not clear what is the main result and contribution.

The empirical studies seem to be very far away from all the terms "gamma-robust features" and "rho-useful features" defined before. It is tested on some contrived test sets that vaguely correlated with the so-called robust/non-robust accuracy. I also don't understand how the empirical studies verify that with both procedures, they first learn "weakly predictive non-robust features". What even is "weakly predictive non-robust features" and how is it reflected in plots?


The biggest problem is the results do not give us constructive ways to improve the training. Even if the paper could give a strong empirical verification that a network overfits due to the fact that it does not learn predictive robust features without proper regularization, it is unclear to me how we could take action to improve the training procedure. The contributions of the paper are not very obvious to me.

---

> ### Author Response · Authors · 2020-11-20
> **Response to AnonReviewer4**
>
> Thanks for the review! Here are our responses to the concerns you raised.
>
> **On the organization and clarity of writing :**
>
> We have made extensive changes to Sections 2, 3 and 4 which we believe will make the paper more precise and easier to understand. We provide a summary of these changes in a reply to the main thread. We have also fixed ambiguities in the writing, making it clear which model/dataset is being referred to in a given context.
>
> **On the connection between the definitions and the experiments :**
>
> We have added a description of the Ilyas et al. $\mathcal{D}_{det}$ experiment in Section 2, which is stated in terms of the provided formal definitions. We use this dataset (and its variants) as the basis of all our further experiments, and we have added explicit references to this setup in Sections 3 and 4.
>
> **On weakly predictive non-robust features :**
>
> The precise definition of a weakly predictive non-robust feature is in the initial part of Section 2. We can infer from the plots that a neural network is learning weakly predictive non-robust features by observing that the “non-robust feature accuracy” is small and non-zero.
> We have also added an illustrative example in Section 2 that provides an example of a weakly predictive non-robust feature, which we hope will help aid intuition.
>
> **On the contributions of the paper :**
>
> It is our contention that understanding the dynamics of deep neural network training is an important problem in itself, even if it doesn’t lead to “constructive ways to improve the training”. This helps us come closer to a principled understanding of why and how neural networks generalize : why we can train large, overparameterized models on relatively few training examples and still expect them to generalize to unseen examples. Our experiments suggest that learning predictive non-robust features is crucial for good generalization, which fills in a small piece of the puzzle.
>
> But aside from this, there are some interesting implications of our two pathway decomposition, one of which we have described in the reply to the main thread.

---

### Official Review · AnonReviewer1 · 2020-10-29
**Interesting contribution but very poorly written**

**Rating:** 5
**Confidence:** 4

**Review:**

This paper studies a novel phenomenon of two different pathways in DNN training that determine whether it learns robust or non-robust features (or both). The paper uses experiments based on adversarial training to validate their hypotheses, and also devise a toy model under which their hypotheses hold.

Pros
- Interesting contribution
- Provides both empirical and theoretical evidence for their claims

Cons
- Very poorly written

Overall, I think the insight the paper is studying is very interesting. Better understanding of the training dynamics of DNNs is an important problem that merits further study, and the authors are taking an important step in this direction. I also appreciate that the authors provide both empirical and theoretical evidence for their claims.

However, this paper is currently poorly written and was very hard to follow. First, their experimental methodology is poorly explained -- e.g., Section 3 and the first paragraph in Section 4.1. They do not use precise terminology, for instance saying they “perturb each example to the next class” (I assume this means they compute an adversarial example for a “7” that is adversarially labelled “8”), and “relabel each adversarial example with its target label (I assume this means label “8”, but I am not 100% sure). Furthermore, they say accuracy on the “shifted labels” corresponds to “robust accuracy”, yet they never explain why this should be the case (again, I think I understand after several re-reads, but the lack of explanation makes it hard to be sure).

Along the same lines, they never explain *why* they are using their methodology. For instance, I’m not sure exactly how Section 3 is connected to Section 4. They say the point is to establish that “a neural network being unable to learn a generalizing model as evidence that there are no highly predictive non-robust features”, but I don’t understand why they need to replicate the results of Nakkiran (2019) to make this claim. Maybe they are replicating to be careful (which is good!), but they never say so. In addition, I also don’t understand this claim very well; e.g., what is the purpose of this connection? Is it to establish their pathways? Section 4 similarly lacks explanation of their methodology.

Finally, it would also be helpful to have an explanation of why these two pathways are interesting. I do think there is some intrinsic value in decomposing different learning pathways in DNNs, and I can imagine several possible implications, but concretely pointing out some implications of these hypotheses would be very helpful.

-------------------------------------------------------------------------------------------------------------------------------

Post rebuttal: I appreciate the improvement in clarity of the paper; however, I think some work remains to improve the clarity of the paper, especially regarding why the two pathways are interesting. I think expanding on potential implications of this mechanism would be very helpful, in addition to the addressing the remaining issues raised by the other reviewers.

---

> ### Author Response · Authors · 2020-11-20
> **Response to AnonReviewer1**
>
> Thanks for the review! Here are our responses to the concerns you raised.
>
> **On the lack of clarity in describing the methodology :**
>
> We have made extensive changes to Sections 2, 3 and 4 which we believe will make the paper more precise and easier to understand. We provide a summary of these changes in a reply to the main thread.
>
> The procedure you described is indeed correct : we would compute an adversarial example for a "7" that is classified by the model as an "8", and relabel it as "8". Figure 3 provides some intuition for why accuracy on shifted labels must come from robust features. There is also a more formal argument in Section 3.
>
> **On the motivation for the experiments in Section 3 :**
>
> Our experiment in Section 3 actually establishes a claim that is **stronger** than that established by the experiments of Nakkiran (2019). We have clarified this, and also rewritten the section such that it builds up to two precise claims. The role of our experiment in establishing these claims has also been clarified.
>
> The final claim (Claim 2) in Section 3 is used again in Section 4 to infer whether a model trained on clean data has learned predictive non-robust features or not. We have added an explicit reference to this in Section 4.
>
> **On why the two pathways are interesting :**
>
> Please see the reply to the main thread.

---

### Author Response · Authors · 2020-11-20
**Addressing common themes in the reviews**

We thank all the reviewers for their detailed reviews. We are glad that most reviewers agreed that the findings of this paper are potentially interesting, and we have made an attempt to present these findings in a more coherent and well-organized manner. Some of the common themes across all the reviews were :

**Clarity and organization of the paper :**

We have extensively modified Sections 2, 3 and 4 such that the narrative is easier to follow and more logically presented. Here is a summary of the major changes :

1. A description of the Ilyas et al. $\mathcal{D}_{det}$ experiment in Section 2. This is stated in terms of the formal definitions for the binary classification setting, and later referenced in Section 3 and 4 to describe the experimental setups.

2. An illustrative example in Section 2 to describe the different types of features. This is also used as a running example in Section 3 and 4 to aid in an intuitive understanding of the experimental setups.

3. Section 3 has been rewritten such that the context for the experiment is clear, and the conclusions from the experiment have been stated as precise claims.

4. A figure in Section 4 that provides the intuition behind why our experiment disentangles robust and non-robust accuracy.

5. The experimental setups in Section 3 and 4 have been stated precisely in terms of the definitions in Section 2, so that there is no ambiguity (eg. about what a “shift” means).

**About Definition 1 :**

We agree with the reviewers that Definition 1 did not provide any insight into understanding the rest of the paper, and indeed it was difficult to leverage it for the same. We have therefore decided to remove it altogether. The important take-away from the discussion in Section 2 is to understand the distinction between the two types of non-robust features, and we believe that the existing discussion suffices for this purpose.

**Why are these pathways interesting? / What are the contributions of this paper? :**

Understanding the dynamics of neural network training is an important problem in itself as it helps us come closer to a principled science of why and how neural networks generalize. However there are some other implications of this two pathway theory too. For instance, adversarial examples generated for other models will not transfer to a Pathway 2 trained model, which means that attackers cannot craft adversarial examples for this model without at least the ability to query the model and observe its outputs for a large number of inputs. We have updated the Conclusion to reflect this insight.

Further, the connection between adversarial transferability that we establish in Section 3 is non-trivial and helps further our understanding of why adversarial examples transfer.

---

### Decision · Program_Chairs · 2021-01-07
**Final Decision**

**Decision:**

Reject

**Comment:**

The paper discusses the dynamics of training neural nets and how they are related to features that are robust and predictive (following Ilyas et al). The reviewers had many comments regarding the presentation of the claim and the validity of the empirical results, as well as their unclear practical implications. The authors have improved the writing somewhat but reviewers still thought the manuscript should be substantially improved so that the claims are clearer and empirical validation is more convincing.
The authors are also encouraged to discuss their results in the context of results on  inductive bias of deep-learning (e.g., results on NTK, rick-regimes, margin maximization etc).